# The New Elastomeric Compounds Made of Butyl Rubber Filled with Phyllosilicates, Characterized by Increased Barrier Properties and Hydrophobicity and Reduced Chemical Degradation

**DOI:** 10.3390/molecules29061306

**Published:** 2024-03-15

**Authors:** Aleksandra Smejda-Krzewicka, Emilia Irzmańska, Konrad Mrozowski, Agnieszka Adamus-Włodarczyk, Natalia Litwicka, Krzysztof Strzelec, Małgorzata I. Szynkowska-Jóźwik

**Affiliations:** 1Institute of Polymer and Dye Technology, Lodz University of Technology, Stefanowskiego 16, 90-537 Lodz, Poland; konrad.mrozowski@dokt.p.lodz.pl (K.M.); krzysztof.strzelec@p.lodz.pl (K.S.); 2Central Institute for Labour Protection—National Research Institute, Department of Personal Protective Equipment, Czerniakowska 16, 00-701 Warsaw, Poland; emirz@ciop.lodz.pl (E.I.); agada@ciop.lodz.pl (A.A.-W.); nalit@ciop.lodz.pl (N.L.); 3Institute of General and Ecological Chemistry, Lodz University of Technology, Zeromskiego 116, 90-924 Lodz, Poland; malgorzata.szynkowska@p.lodz.pl

**Keywords:** butyl rubber, phyllosilicate, barrier properties, hydrophobicity, chemical degradation

## Abstract

The aim of the study was to produce new elastomeric materials containing butyl rubber (IIR) filled with silica and phyllosilicates (vermiculite, montmorillonite, perlite or halloysite tubes) with enhanced hydrophobicity and barrier properties and reduced chemical degradation. It was found that the filler type had a significant impact on the degree of cross-linking of butyl rubber and the properties of its vulcanizates. The highest degree of cross-linking and the highest mechanical strength were achieved for IIR composites filled with Arsil with perlite or halloysite tubes. The highest surface hydrophobicity (119°) was confirmed for the IIR vulcanizates with Arsil and montmorillonite. All tested samples showed high barrier properties because both the gas diffusion rate coefficient and the permeability coefficient reached low values. Both unfilled and filled IIR vulcanizates retained chemical resistance in contact with methanol for 480 min. Hour-long contact of a polar solvent (methanol) with each of the vulcanizates did not cause material degradation, while the presence of a non-polar solvent (n-heptane) worsened the mechanical parameters by up to 80%. However, the presence of fillers reduced the chemical degradation of vulcanizates (in the case of cured IIR filled with Arsil and halloysite tubes by 40% compared to the composite without fillers).

## 1. Introduction

Rubbers have become essential materials in the industry with the advancement of plastic technology. The popularity of rubbers is due to good performance properties and the possibility of functionalization [1]. The areas of particular interest in elastomer research include such features as self-healing, shape memory effect [2,3], superhydrophobicity [4], reinforced mechanical properties, improved barrier properties [5], and enhanced damping properties [6,7].

The opportunities provided by rubbers make it possible to create both elastomeric materials that perfectly dampen vibrations and those that can be repaired after breaking. For this reason, elastomers are widely used in such fields as automotive, aerospace, industrial equipment, and household appliances [2,3,4,7,8,9]. Noteworthy among rubbers is butyl rubber (IIR). It is a synthetic elastomer that is formed by the polymerization of isobutylene and isoprene. IIR vulcanizates exhibit very high anti-aging resistance and limited permeability to gases and liquids. In addition, they are resistant to weathering, ozone, hot air, acids, and bases. Butyl rubber is mainly used for products with reduced permeability to gases and liquids such as tire inner tubes, hoses, gaskets, and membranes [10,11,12]. However, the latest research in the area of butyl rubber application shows that its use could be much wider.

El-Sabbagh S.H. et al. [13] investigated the compatibility of natural rubber (NR) with butyl rubber to achieve elastomeric blends featuring both IIR and NR properties. The authors showed that macrophase separations form in the IIR/NR mix, which is disadvantageous for the properties of such blends. Moreover, a compatibilizer was used, which decreased the extent of phase separation and increased interfacial adhesion between phases, consequently reducing the size of dispersed phase domains, which led to better mechanical properties including tensile strength and a higher value of Young′s modulus. Sukharev N. et al. [14] studied elastomeric blends containing butyl rubber and ethylene-propylene-diene rubber (EPDM). The study determined the influence of the proportion of elastomers in uncured blends and their effect on phase structure formation patterns in multicomponent polymer systems, changes in molecular mobility, and ozone resistance. The research has shown that there is such a ratio of EPDM to IIR to obtain the highest possible resistance to ozone. These materials can be successfully considered for use in applications where ozone concentrations are higher, i.e., at heights above 15 km or in the aerospace industry. A novel strategy to prepare interpenetrating polymer networks (IPNs) based on butyl rubber and poly(n-octadecyl acrylate) (PC18A) was developed by Tavsanli B. et al. [15]. Solvent-free UV polymerization of the n-octadecyl acrylate monomer in the IIR melt at ambient temperature resulted in IPNs with self-healing and shape memory functions.

According to Guo X. et al. [16], butyl rubber can be applied to materials with high damping electromagnetic interference or multi-absorbing materials, including absorbing electromagnetic waves. The study showed that the composites of butyl rubber with single-walled carbon nanotubes (IIR/SWCNT) achieved good mechanical performance (tensile strength reached 15 MPa), and the total electromagnetic shielding efficiency of the material increased to 23.8 dB. In addition, the authors showed that water-induced modification of the composite achieved good dispersion of SWCNTs to enhance electromagnetic shielding while maintaining a wide damping temperature range from −55 °C to 40 °C with a damping factor above 0.2. Chameswary J. et al. [17] studied butyl rubber filled with barium titanate (BaTiO_3_) with micro- or nanometric sizes. The research showed that such composites exhibited good mechanical properties, and they are flexible and absorb radio frequency vibrations. These results prove that these composites are proper candidates for the core of flexible dielectric waveguides and applications in flexible microwave substrates. Hao, S., et al. [18] tested butyl rubber filled with graphene. Studies have shown that the proper formulation of such a composite makes it possible to achieve excellent mechanical properties, high conductivity, and high barrier to water vapor. The conductivity of the IIR/graphene nanocompound at a graphene content of 3.76% prepared by Liquid Phase Redispersion reached more than seven orders of magnitude higher than the conventional twin-roll mixing method. Due to the existence of homogeneously distributed networks of segregated graphene, the tensile strength and elongation at break for the IIR/graphene nanocomposites increased by 410 and 126%, respectively, at a graphene content of 3.76%. The IIR/graphene vulcanizate exhibited such electrical properties that it can be used as a wearable sensor and physiological signal detection.

The presented examples of filled IIR compositions show the advantages of using innovative substances such as elastomer fillers. However, for decades, the rubber industry, especially the tire industry, has mainly used carbon black as a reinforcing filler. Since its structure and surface can be changed over a wide range, carbon black can meet a wide variety of requirements. However, with the passing of time and the necessity to invent alternatives to carbon black, various types of silicas have been developed. In the 1940s and 1950s, the specific surface area and structure of silica were constantly adjusted to satisfy new requirements in the field of rubber materials [19,20,21]. As the development proceeded, the advantages and disadvantages of silica compared to carbon black began to be perceived. Therefore, the research began for new and more easily available fillers for rubber, which will not cause as much carbon dioxide emissions as carbon black. The first and most obvious reason for using fillers is to improve mechanical properties, that is, to enhance the elastomer [19,22]. Fillers such as carbon black or silica can reinforce cured rubber, improving its tensile strength, durability, and wear resistance. This is particularly important in applications where the rubber products will be subjected to significant stress or abrasion. In addition to conventional fillers, various types of synthetic or natural substances are used, which also reinforce the composite by forming elastomer–filler and filler–filler interactions, thus creating a spatial network in the structure of the elastomer, which is responsible for carrying stresses [19,21,22]. Another reason for using fillers is to reduce the cost of the rubber product. Fillers then act as diluents and reduce the amount of expensive rubber in the product. This is important to keep the right properties reducing production costs. Another reason is to control physical properties; fillers are used to adjust hardness, elasticity, and electrical and thermal conductivity. This allows manufacturers to customize rubber to suit specific requirements. In addition, the use of fillers allows the dimensional stability of rubber composites to be maintained. The incorporation of a filler into an elastomer matrix reduces shrinkage and increases the ability to retain shape over time. This is important in terms of providing the accuracy of manufactured parts, the efficiency of seals and gaskets, and the overall performance and reliability of rubber components in various applications [20,21,22]. Additionally, some fillers improve the processability of rubber compounds by facilitating molding, extrusion, or other processing during production. The above reasons have contributed to the search for various fillers with unique properties, and the specific choice of filler will depend on the requirements of the product and the manufacturing process. Therefore, in recent years, more and more research has been conducted on alternatives to silica and carbon black [19,20,21,22,23]. Substitution of standard reinforcing fillers with montmorillonite (MMT) has improved some of the desired properties of elastomeric compounds in addition to some reductions of the final weight and price of the final products [24]. Montmorillonite organoclay has also been used as a filler in the IIR composites to improve their physical and mechanical properties [25]. In this study, melt mixing in an internal mixer was selected as the method of dispersing organoclay in the elastomeric matrix, and the intercalation of rubber chains into the clay gallery was deduced from the increase in basal spacing of the silicate layers as was measured by XRD. The highest basal spacing was detected for the amount of MMT equal to 3 phr. Dispersion and distribution of the organoclay were observed by SEM. The authors proved that organoclay content and structure had a large impact on the mechanical and rheological properties of nanocomposites as well as the permeability of carbon dioxide gas through their films.

This study has focused on layered silicates and their effects on butyl rubber vulcanizates because it is a very promising group of compounds, that indicate suitable properties in the case of novel elastomeric materials. In this paper, the main focus was on the hydrophobic and barrier properties of butyl rubber vulcanizates filled with various phyllosilicates. A material whose surface exhibits hydrophobic or superhydrophobic properties is the target of much scientific research [4]. The term “superhydrophobic” describes surfaces with a Young′s angle of more than 150° and indicates very low wettability. This angle is defined as the mechanical equilibrium of the drop under the action of three interfacial tensions: solid–vapor, γ_sv_; solid–liquid, γ_sl_; and liquid–vapor, γ_lv_ [26,27,28]. This equilibrium relation is known as Young′s Equation (1):(1)$$\gamma_{\mathrm{lv}}{\cos\theta }_{Y}={\;\gamma }_{\mathrm{sv}}-\gamma_{\mathrm{sl}}$$
where γ_lv_, γ_sv_, and γ_sl_ represent the liquid–vapor, solid–vapor, and solid–liquid interfacial tensions, respectively, and θ_Y_ is the Young′s contact angle [28].

The popularity of such surfaces was started by the “lotus effect” described by Barthlott and Neinhus [29,30]. The lotus leaves owe their unusually high contact angle to a characteristic structure that scientists want to replicate for superhydrophobic materials. Examples of applications for such products could include self-cleaning materials [31], anti-corrosion [32], anti-icing [32,33], protective coatings, and impermeability to liquids.

Hydrophobicity microscopically is the tendency of chemical molecules to repel water molecules from each other, while hydrophobicity macroscopically is the property of a material’s surface to be non-wettable by water. Associated with this phenomenon in polymeric materials are such terms as surface tension, surface free energy, and contact angle [34]. The wettability of polymer surfaces, including elastomers, depends on the contact angle, e.g., the angle formed by the tangent to the surface of a droplet on the surface of a solid, and it is highly dependent on the surface tension [35,36,37].

The various values of Young′s angle lead to different phenomena at the solid–liquid interface. Therefore, several values of ϴ are distinguished as follows:−ϴ = 0, when the liquid wets the surface of the solid and tends to spontaneously spread on it; this state is defined as the critical value of the surface tension of the liquid;−0 < ϴ < π/2, when the liquid spills on the surface and tends to shrink on the surface within a limited range;−ϴ > π/2, when the liquid does not spread on the surface and tends to shrink on the surface of the solid, forming droplets [35,36,37].

In the context of the water–polymer interface, it must be considered that the contact angle is a measure of the hydrophilicity of the surface of a polymer material. Because water exhibits the highest surface tension compared to other liquids, the character of the polymer surface can be determined by the value of the contact angle. A hydrophilic surface is distinguished by the fact that the applied droplet spreads, thereby wetting the surface (the angle ϴ is small), for example, it is worth mentioning the surfaces of non-oxidized metal or glass (ϴ < 20°). On the opposite side, according to the literature, a hydrophobic surface is characterized by the fact that the applied water droplet does not spread on the polymer surface (the angle ϴ is large), such as the surface of polypropylene (ϴ = 110°) or polystyrene (ϴ > 97°) [34].

Other important parameters describing elastomeric materials are their barrier properties. These are the features that determine whether a type of polymer can be used as, for example, a film or protective coating against a toxic substance. Three coefficients are used to describe them: permeability coefficient, diffusion coefficient, and solubility coefficient [5,38,39]. To describe the permeation of gases through an elastomer, a diffusion mechanism is used, which occurs across the material due to a pressure gradient. In this case, the constant volume and variable pressure method is used to measure this coefficient. A vacuum was applied to both sides of the polymeric material, and the permeability coefficient P is described by the following Formula (2):(2)$$P=\frac{V\cdot t_{f}}{A\cdot R\cdot T\cdot \Delta_{P}}\cdot \left(\frac{\mathrm{dp}}{\mathrm{dt}}\right)$$
where V is the total amount of gas permeation through the sample into a cell, t_f_ is the sample thickness, A is the sample area, R is the gas constant, T is the absolute temperature, Δp is the pressure gradient across the sample, dp/dt is the transmission rate [5,38,39].

In addition, the permeability coefficient (P) combines the effects of both the diffusion coefficient (D) and the solubility coefficient (S) and can be explained as Formula (3):P = D∙S(3)
where D is the diffusion coefficient, which describes the kinetic aspect of transport, S is the solubility coefficient, which is related to the affinity of the penetrating substance (gas or liquid).

The above relation holds true when the value of D is independent of concentration and the value of S follows Henry’s law. It is used to describe gas transport in polymeric composites reinforced with impermeable nanofillers. In the above diffusion–solubility model, penetrant molecules initially dissolve into the high-pressure face of a film then diffuse across it through thickness and finally desorb at the low-pressure face. Thus, the permeability of a penetrant depends on both its diffusivity and solubility. These properties can be changed by the molecular structure of the polymer and environmental factors [38,40].

To increase the barrier properties of elastomers, layered fillers such as montmorillonite, graphene, vermiculite, or halloysite are introduced [39,40,41,42,43,44,45]. The layers of this filler must “split” and form a difficult path for the substance to penetrate the material under investigation. This process is called exfoliation and is not simple to achieve when the filler is incorporated in situ into the polymer. Studies show that the most effective way to stop gas diffusion is to position the filler layers in an orientation perpendicular to the movement of the penetrating gas and increase the interactions at the polymer–filler interface [41,42,43,46].

It is common knowledge that the morphology and dispersion of layered aluminosilicates are key factors affecting the barrier properties of the composite. Therefore, proper dispersion and achieving a high degree of exfoliation of the layered nanofiller in the polymer matrix are the most important challenges in producing nanocomposites with improved barrier properties. According to the literature, there are three possible morphologies in nanocomposites (Figure 1) [38,47,48]:−Tactoid (conventional composite), when there is no separation of layered filler packages, micrometer-sized structures are present in the polymer medium;−Intercalated nanocomposite, when the polymer is located between parallel filler galleries, separation of filler layers occurs;−Exfoliated nanocomposite, when the structure with the highest degree of dispersion of the filler in the polymer matrix, the polymer chains cause separation of the filler layers [49,50,51].

The study aimed to produce new elastomeric materials containing butyl rubber and characterized them by enhanced hydrophobicity and barrier properties and reduced chemical degradation. To test the research, six various rubber compositions were prepared, which differed in the kind of filler. Phyllosilicates were chosen as fillers in the tested IIR compositions because they have the ability to interact with organic matter [49,50,51,52,53,54,55], as well as the tendency to separate the packages that constitute them into nanometer-thick layers, which theoretically allows the obtaining of materials classified as nanocomposites [56,57,58,59,60,61].

## 2. Results and Discussion

### 2.1. Characteristics of Phyllosilicates

Silicates, i.e., natural minerals, are most often formed as a result of the transformation of dust or volcanic rocks. Phyllosilicates have different structures. Two-layer silicates with a 1:1 structure are important for the rubber industry. This group mainly includes kaolinite, halloysite, and perlite. These compounds have unique properties and after adding them to a polymer matrix, they can form nanocomposites by intercalating the packages [62,63]. Halloysite (HNT) used in the tested IIR compositions is a mineral characterized by high porosity and specific surface (which is very desirable for elastomer fillers), high ion exchange (ability to absorb heavy metals), and easy chemical and mechanical processing [64]. It is an aluminosilicate containing approx. 45% silica and approx. 40% aluminum oxide. The remaining ingredients are water and trace amounts of metal oxides, i.e., TiO_2_, Al_2_O_3_, FeO, MgO, CaO, Na_2_O, and K_2_O. Halloysite usually forms very small, tubular crystals visible under very high magnification. The mechanical, thermal, rheological, and barrier properties of HNT-modified polymers depend on the two most important factors: the degree of particle dispersion and the compatibility of the polymer with the nanofiller. Pearlite (PER) is a two-phased, lamellar compound composed of alternating layers of ferrite (87.5%) and cementite (12.5%). Its chemical formula depends on the rock compositions, but perlite consists of the following metal oxides: SiO_2_ (71–75%), Al_2_O_3_ (12–18%), Na_2_O (3–4%), K_2_O (4–5%), Fe_2_O_3_ (0.5–1%), and MgO (0.1–1.5%). It is most often used as a filler for polyethylene, polypropylene, or poly(vinyl chloride) [65].

Three-layer silicates include compounds such as montmorillonite, mica, talc, and vermiculite. Silicates with three-layer packages with a 2:1 structure type are distinguished by the fact that the octahedral layer is located between two tetrahedral layers with their vertices facing each other [62]. As the most famous of this group of silicates, montmorillonite (MMT) is a nanofiller. It is a mineral from the group of dioctahedral smectites with the chemical formula (Al_4−x_Mg_x_)[Si_8_O_20_](OH)_4_, where x = 0.67 and the Si:Al ratio is approximately 5:2. Another silicate with a three-layer structure is vermiculite (VER) [66]. It is a phyllosilicate that has a layered structure composed of an octahedral layer and two tetrahedral layers. The chemical composition of this compound is (Mg,Fe,Al)_3_[(Al,Si)_4_O_10_](OH)_2_·4H_2_O [62]. During the heating, the layers expand, increasing the volume of the mineral. Vermiculite is light, inexpensive, non-toxic, chemically inert, and resistant to thermal decomposition. These properties make this material ideal as an insulator or filler in insulating applications [62,63,66,67,68,69,70].

The filler structure was examined based on images acquired with a scanning electron microscope (SEM), with sample pictures presented in Figure 2.

The Arsil (ARS) particles are finely aggregated and agglomerated (Figure 2a). The structure of montmorillonite (Figure 2c) and halloysite tubes (Figure 2e) looks very similar. The vermiculite particle grains are completely different from the other fillers used and they form large and flat grains, and there are many empty areas between them (Figure 2b). The perlite particles have strongly jagged shapes and are loosely bound together (Figure 2d).

### 2.2. The Influence of Phyllosilicates on the Course of IIR Cross-Linking

The recipe for the compounding of the IIR composites filled with phyllosilicates is given in Table 1.

The vulcanization parameters of rubber mixes depend on several factors, for example the type of rubber, the cross-linking agent, and the filler type. The minimal rheometric torque (T_min_) is responsible for the viscosity of the mix, while the rheometric torque increment (ΔT) corresponds to the degree of cross-linking of butyl rubber. The next parameters are the scorch time (t_02_) and optimal vulcanization time (t_90_), which determine how long the mix will be vulcanized. The results in Table 2 clearly show that the addition of montmorillonite increased the vulcanization time (16.0 min), while the sample filled with vermiculite achieved the shortest cross-linking time (9.4 min). The other filled composites achieved similar t_90_ values (from 10.3 to 12.5 min) compared to the unfilled IIR. The addition of phyllosilicates resulted in shorter scorch times, as each filled sample reached a lower t_02_ value than the reference sample, which proves that the filling of butyl rubber is related to a reduced time for safe processing.

The filled samples achieved a higher viscosity than the unfilled sample, which is shown in Table 2 by comparing the T_min_ values (IIR: 0.66 dNm; IIR/VER: 1.06 dNm). Moreover, each of the filled samples achieved a rheometric torque increment equal to or higher than the unfilled compound. This indicates that added phyllosilicates cause an increase in the viscosity of the elastomer and an increase in the degree of cross-linking. The IIR/PER and IIR/HNT samples recorded a higher increase in rheometric torque (3.87 dNm and 3.93 dNm, respectively). The MMT-filled IIR sample reached the lowest ΔT_max_ (2.37 dNm), while its minimal rheometric torque was the largest (1.34 dNm). The degree of conversion (e.g., degree of transformation of plastic material into elastic material) calculated on the basis of vulcametric results also achieved the lowest result (0.84) for the sample containing MMT. The α_c_ value for the remaining IIR compositions was comparable and higher by 14% (α = 0.95). These results indicate a lower degree of cross-linking of butyl rubber if it was filled with montmorillonite. Table 2 also summarizes the CRI values for all composites, so it is observed that the addition of MMT or HNT significantly lowered the CRI values (7.5 and 8.8 min^−1^, respectively) relative to the refereed compound, whereas for the other samples, this parameter was like the IIR sample.

Figure 3 shows the time relationship of the torque for all samples. It is noted that the course of the curves for all filled samples (except MMT) differed at the beginning of the cross-linking process in relation to the unfilled IIR. On the curves of the IIR and IIR/MMT samples, it is not noticed that the torque decrease during the scorch time was so pronounced; moreover, it was longer than for the other mixes. In addition, these two compositions were characterized by the least torque increment. The other curves were very similar; only curves 3 and 6 could be differentiated due to their scorch times, which were the shortest among the tested compositions. According to the data in Table 2, Figure 3 also shows that IIR/PER and IIR/HNT composites achieved the highest values of rheometric torque increment. All curves regardless of whether the sample was filled with phyllosilicates showed no reversion ability.

Besides the vulcametric parameters, the degree of cross-linking of vulcanizates could be determined by the equilibrium swelling. For this purpose, several parameters were determined. The first is the volumetric equilibrium swelling (Q_v_), which accounts for how much a given vulcanizate is cross-linked. The next is the content of the eluted fraction during the swelling (−Q_w_), which determines how much substance has been eluted after contact with a selected solvent. The volume fraction of rubber in swollen material (V_R_) determines the degree of cross-linking of the vulcanizate; the higher the value of this parameter indicates the greater the degree of cross-linking (α_c_). Table 2 shows the results of equilibrium swelling and states that the addition of Arsil or Arsil with halloysite tubes or perlite improves the degree of cross-linking of the IIR vulcanizates. The values of α_c_ for these samples are 0.38–0.39, while this parameter for the reference sample was 0.28. Thus, it was observed that there was a significant increase in the degree of cross-linking by filling the vulcanizates. The Q_v_ values obtained for all filled samples (without IIR/MMT) prove that the addition of phyllosilicates increased the degree of cross-linking. The IIR/ARS and IIR/HNT samples obtained the smallest Q_v_ value (2.58 mL/mL), which indicates the highest degree of cross-linking. The contents of the eluted fraction, except for the IIR/ARS sample (−Q_w_ = 0.14 mg/mg), were at a similar level of values to the reference sample, i.e., about 0.27 mg/mg. The volume fraction of rubber in the swollen material also confirmed that the samples containing ARS, PER, and HNT had the highest degree of cross-linking, as the V_R_ values were the largest among the samples tested (IIR/ARS: 0.280). It is worth emphasizing that the equilibrium swelling results of the produced IIR vulcanizates presented in this part correlate well with the vulcametric results.

### 2.3. The Influence of Phyllosilicates on the Mechanical Properties of IIR Vulcanizates

The results of the tensile strength test before and after thermo-oxidative aging are summarized in Table 3. The obtained results before aging show that the addition of Arsil or Arsil with perlite or halloysite tubes slightly increased the stiffness of the samples. The composite samples filled with vermiculite and montmorillonite reached comparable S_e100_ values as the reference sample. In Table 3, TS_b_ results confirmed the fact that the addition of phyllosilicates to IIR mix increased the mechanical strength of the vulcanizate. Each of the tested composites increased its tensile strength value by a minimum of 50%. Such a correlation might be due to the formation of a spatial network of bonds between the filler in the elastomer matrix, which results in the rigidity and enhancement of the elastomeric composite. The greatest TS_b_ values were obtained by the IIR/HNT (13.4 MPa), IIR/ARS (11.1 MPa), and IIR/PER (11.0 MPa) vulcanizates, which indicates the best mechanical strength of these two compositions. The elongation at break for all filled samples is more than twice that of the unfilled IIR vulcanizate (IIR: 590%, IIR/PER: 1289%). However, in addition to the strength parameters before aging, Table 3 summarizes the results after aging. This is important because as time passes, the mechanical properties of rubber products undergo various changes, because of which the products may become useless due to increased stiffness and cracking. Among the factors influencing the degradation of elastomer properties, oxygen and increased temperature are the most important. The results in Table 3 show that after thermo-oxidative aging, the tested composite became more rigid, which could be seen to a greater degree for the IIR/HNT, IIR/PER, and IIR/VER samples. The values of S_e100_ and S’_e100_ for the IIR/VER vulcanizate were 0.48 MPa and 0.66 MPa, respectively, and this was since under the influence of increased temperature and the presence of oxygen the vulcanizate begins to cross-link, making it more rigid and resistant to deformation. This relationship was visible for silicate-filled composites, while for the unfilled vulcanizates, no major changes in strain stress can be observed. The tensile strength after aging increased for the IIR/MMT and IIR/VER samples (5.7 MPa and 8.7 MPa, respectively), while it decreased for the other compositions. The IIR/PER vulcanizate recorded the greatest decrease in tensile strength among those tested (from 11.0 MPa to 6.6 MPa), while the others decreased by a few tenths of an MPa. After aging, a decrease in elongation values at break was also observed for all vulcanizates. The filled samples, however, continued to reach values greater than the referenced composition. The filled vulcanizate with the highest resistance to thermo-oxidative aging was the IIR/MMT composition, as it achieved an aging factor of AF = 0.67, closely behind was the IIR/VER vulcanizate with a value of AF = 0.63. These values can be explained by the fact that the two samples have the lowest degree of cross-linking and after thermo-oxidative aging, the vulcanizates undergo a process of cross-linking, while the other vulcanizates undergo a process of degradation due to the higher degree of cross-linking.

Other mechanical properties are summarized in Table 4. The results of the hardness test of the vulcanizates showed that the addition of phyllosilicates increased hardness. The IIR/PER and IIR/HNT samples achieved the highest Shore A hardness values (57.6 °ShA and 56.2 °ShA, respectively). The IIR/MMT vulcanizate, despite the weakest mechanical strength among the filled compositions, showed the highest tear resistance (T_s_ = 5.46 N/mm); moreover, the addition of silicates made this parameter improved for each sample relative to the unfilled sample. To determine the damping properties of the rubber, hysteresis loops were determined on a graph at five tensile cycles and the Mullins effect was calculated. This test was performed only for filled composites, due to the study of filler–elastomer interactions.

In Table 4, the results of hysteresis losses during the first and five stretching cycles show that the IIR/HNT sample recorded the highest losses (ΔW_1_ = 51.2 N·mm and ΔW_5_ = 29.9 N·mm). The composite that achieved the fewest losses is the IIR/ARS vulcanizate (ΔW_1_ = 39.9 N·mm and ΔW_5_ = 27.4 N·mm), so it is considered the best damping composite among those tested. The value of the Mullins effect (E_M_) shows that the smaller filler dispersion was better in the elastomeric matrix or indicates small agglomerates. The IIR/ARS composite had the smallest value of E_M_ (13.5%), which determines that the filler in this sample was well dispersed.

### 2.4. The Influence of Phyllosilicates on the Dynamic Properties of IIR Vulcanizates

The dynamic properties of rubber materials depend mainly on the quantity and activity of the filler used. The distribution of filler particles in the elastomeric matrix is also an important factor. Table 5 compares the results of the dynamic properties of the filled vulcanizates tested. The unfilled IIR composite was not considered in this study. The highest value of the storage modulus G′ was achieved by the IIR/PER composite (0.278 MPa), indicating the greatest resistance to deformation of this sample and the greatest filler–filler interactions, which cause reinforcement of the composite. The other vulcanizates obtained smaller values of the storage modulus. The loss modulus represents the viscous part, or the amount of energy dissipated in the sample, and is related to the material’s ability to dissipate stress through heat. The IIR/ARS sample (G″_max_ = 0.012 MPa) dissipates energies through the heating and shows the best damping properties due to the smallest value of G″, which agrees with the study of damping properties by hysteresis loops and the Mullins effect.

The largest Payne effect values were achieved by the IIR/PER and IIR/MMT samples (0.256 MPa and 0.181 MPa, respectively). Therefore, these vulcanizates probably had larger aggregates and the spaces between them were smaller than the others, which was also confirmed by the SEM analysis (Figure 2). The IIR/ARS composite probably had the smallest aggregates due to the smallest value of ΔG′ = 0.040 MPa. In addition, each sample had a storage modulus greater than the loss modulus, which was why these materials were mainly elastic.

The variation of the storage modulus from the increased oscillation strain is shown in Figure 4. The IIR/PER vulcanizate (curve 4) had the highest stiffness, and therefore the greatest value of the storage modulus. The addition of perlite caused the formation of filler–filler bonds in the elastomer matrix, which made the G′ value increase. The strong Payne effect in this case was probably caused by the structure of pearlite, and its particles are highly corrugated (Figure 2d), increasing its specific surface area and interactions with butyl rubber macromolecules. The IIR/MMT sample (curve 3) also showed a high value of G′ (0.183 MPa), indicating the appearance of a large amount of filler–filler bonds. As the oscillation strain increased, the storage modulus for each of the tested compositions decreased. After crossing 10%, a noticeable decrease in the value of the storage modulus was observed for each curve in Figure 4. In addition, it was observed that as the deformation increased, the elastic properties of the tested vulcanizates decreased, which was due to the deterioration of the spatial network structures of the filler in the elastomer matrix.

Figure 5 shows the variation of loss modulus with increasing oscillation strain. Loss modulus measures the energy dissipated or lost as heat per cycle of sinusoidal deformation. The loss value was the largest for the IIR/PER sample (curve 4), the other compositions achieved a lower sinusoidal peak value. In addition, a peak shift was observed with the IIR/HNT vulcanizate (curve 5). This sample had its peak at about a 60% oscillation strain, while the other compositions had this peak at about a 5% oscillation strain. The IIR/ARS vulcanizate (curve 1) in the graph shown did not have a peak at all, indicating little energy loss through heat. The shift of the peak perhaps indicated specific interactions between the filler and the elastomeric matrix in the IIR/HNT sample.

### 2.5. The Influence of Phyllosilicates on the Hydrophobicity of IIR Vulcanizates

The contact angle is aimed at determining the nature of the surface of the polymer under test. The surface can be hydrophobic or hydrophilic. For polymers, the addition of selected fillers is often important, as they allow the surface to remain hydrophobic. In Figure 6 the results of the contact angle are summarized. It is noted that filled samples obtain greater hydrophobicity than the unfilled IIR vulcanizate. Also, the addition of phyllosilicates improved the hydrophobicity of butyl rubber materials. The largest contact angle was obtained by the IIR/MMT composition (Φ = 118.9°). The other filled samples also obtained such a value of Φ that they are considered materials with improved hydrophobicity. Among the filled products, the IIR/VER vulcanizate had the lowest contact angle value, which was related to the poor dispersion of vermiculite in the elastomeric matrix. When the fillers are unequally distributed in the matrix, the agglomerates and aggregates are formed, which causes disturbances in the character of the composite surface. The presented results clearly show that IIR/HNT (Φ = 116.2°) and IIR/PER (Φ = 110.4°) vulcanizates had better dispersion filler than the IIR/VER composition. In addition, an important aspect influencing the hydrophobicity of the rubber material is the shape and size of the filler particles. According to Figure 2, it is important to note that vermiculite has the largest particles with regular shapes among the tested fillers. According to [71,72,73,74,75], fillers, which have a smaller particle size (nano), cause an increase in the roughness of the material’s surface and this results in an increase in the hydrophobicity of the composite according to the Wenzel model and Cassie–Baxter model [72,76,77]. Thus, the IIR/MMT, IIR/PER, and IIR/HNT samples achieved higher contact angle values than the other samples, although all fillers belong to the hydrophilic. In summary, the degree of dispersion, the degree of surface roughness, and the choice of filler and its size also affect the hydrophobicity of the composite [71,72,73]. Nanofillers cause greater surface roughness, but it is more difficult to disperse them evenly in the elastomeric matrix; therefore, the selection of appropriate components when creating new rubber materials is very important [74,75].

### 2.6. The Influence of Phyllosilicates on the Barrier Properties of IIR Vulcanizates

The barrier test was conducted to determine the effect of added fillers on air permeability. The calculated gas transmission rate and permeability coefficients are summarized in Table 6. Analyzing the results, it was noted that the addition of fillers increased both GTR and P values, making the filled composites more permeable than the reference sample. The IIR/PER and IIR/HNT vulcanizates were the least permeable, as they achieved the lowest GTR and P values. Filled vulcanizates should show lower permeability than the unfilled sample, but the problem could be the lack of modification of the fillers or the process of incorporating them into the elastomeric matrix. An important issue in achieving increased barrier properties is to provide the right morphology of the composite (torturous diffusion path). Aiming for exfoliation of filler layers is essential to raise the barrier properties of the product. In this case, it is possible to achieve a very low degree of exfoliation of layered filler packages, which translated into an unsatisfactory result.

### 2.7. Resistance to Permeation by Liquid Chemical Substances

Resistance to permeation by the test chemical was determined by measuring the breakthrough detection time. Breakthrough detection time is the elapsed time from the start of the test to the time the test chemical first breaks through the barrier material and can be measured in the collecting medium. Resistance to permeation by liquid chemicals was conducted according to the procedures described in the Materials and Methods section, with the results given in Table 7. The tests involved two substances: methanol, which is a polar solvent, and n-heptane, which is non-polar.

Previous research [78] indicates interest in the effects of fillers on the gas permeability of polymeric materials. In their work, Takahashi et al. observed that the addition of 30 wt% vermiculite as a nanofiller to butyl rubber caused a more than 20-fold decrease in the permeability of helium, hydrogen, oxygen, nitrogen, methane, and carbon dioxide through the resulting vulcanizates as well as a higher diffusion coefficient for those gases (by two orders of magnitude) as compared to IIR without the filler. While the cited study revealed a beneficial (reducing) effect of nanofillers on the gas permeation rate through elastomeric materials, these findings cannot be directly extrapolated to liquid permeation due to differences in the size and shape of gas and liquid molecules. In the present study, the barrier properties of non-polar IIR were found to depend on solvent polarity as well as on the surface properties of the nanofillers added to the system.

In the case of a polar solvent, the addition of fillers does not change the resistance to penetration of liquid chemicals. The sample retained chemical resistance in contact with methanol for 480 min, and in all cases, the highest performance level of 6 against the permeation of liquid chemicals has been achieved by the EN ISO 374-1:2016 standard [79]. The presence of fillers did not reduce the protective parameter.

Fillers had no effect on the permeation of a polar solvent (methanol) through the non-polar material (IIR), which exhibits a greater thermodynamic affinity to IIR than n-heptane. In the case of n-heptane, the vulcanizates showed lower resistance than for contact with the polar solvent.

Differences in the time taken for the liquid chemical substance to penetrate and change were observed depending on the filler addition and times in the range from 210 to 380 min were recorded. The lowest value of chemical resistance was observed for the IIR/VER vulcanizate, in which the tested material achieved a performance level of 4. In other cases, an effectiveness level of 5 was achieved by the EN ISO 374-1:2016 standard, and the longest permeation resistance time of 380 min was obtained for the IIR/MMT sample.

While the n-heptane breakthrough time for unfilled IIR was 249 min, it increased by approx. 30–45 min (20%) for the IIR/ARS, IIR/PER, and IIR/HNT composites (without significant differences between nanofillers used). However, the breakthrough time for the IIR/MMT sample was as long as 380 min, which means a 50% increase as compared to the unfilled composite. The differential effects of the studied nanofillers on breakthrough time for IIR with respect to liquids of varying degrees of thermodynamic affinity are probably related both to differences in liquid sorbability and to the heterogeneous distribution of surface energy in fillers with dissimilar chemical structures. Some influence on the barrier properties of IIR composites may also be exerted by differences (if any) in the degree of dispersion and aggregation of nanofiller particles [80].

Differences in breakthrough time for composites with nanofillers may result from different amounts of n-heptane retained in the nano- and mesopores of aggregates of filler particles (with the fillers differing in the surface polarity of their solid state particles). Differences in polar liquid breakthrough times for composites also depend on the liquid desorption rate, which is associated with the immobilization of its particles in aggregates of nano- and mesopores and with different forces bonding the nonpolar liquid with the surfaces of the applied fillers of different polarities. Among the studied fillers, the highest polarity was found for MMT, which also exhibited the highest chemical resistance when challenged with a non-polar solvent [81].

### 2.8. Chemical Degradation

Chemical degradation due to the action of liquid substances is defined as the percentage change in mechanical properties, i.e., piercing force, as a result of contact with chemical agents. Chemical degradation tests for the studied vulcanizates were conducted according to the procedures described in the Materials and Methods section, with the results given in Table 8. As a result of the analysis, it is concluded that the polar solvent does not degrade any of the tested vulcanizates. The opposite effect is present when vulcanizates come into contact with a non-polar solvent, i.e., n-heptane. It is observed that the addition of nano-fillers weakens the chemical degradation process, compared to the pure vulcanizate, by up to 40%. The IIR/ARS and IIR/MMT as well as IIR/VER and IIR/PER composites achieved similar chemical degradation values.

According to the literature data, cured butyl rubber is characterized by a similar resistance to chemicals, including solvents and mineral oils, as silicone rubber. However, as compared to butyl rubber, the sorptive capacity of silicone is almost 40% greater for non-polar solvents (e.g., benzene) and much higher for polar chemicals, such as ethanol (leading to a mass increase by 15 units after 168 h of immersion in the solvent). Consequently, the current findings concerning chemical degradation and resistance to liquid permeation indicate an improvement in the barrier properties of filled vulcanizates (e.g., IIR/ARS) as compared to non-polar ones (IIR). Szadkowki et al. also reported an improvement in barrier properties by the addition of materials such as perlite and vermiculite to the polymeric matrix, especially at high concentrations, when mineral particles may form an impermeable layer due to their structure [82]. Vulcanizates filled with VER and PER, with a finer structure, exhibit a higher mechanical strength following a controlled process of immersing a vulcanizate surface in a non-polar solvent (i.e., n-heptane). The other fillers with a looser and finer structure, as well as those with similar structures, are characterized by even higher barrier properties, with permeation resistance results indicating longer n-heptane breakthrough times for the IIR/MMT and IIR/HNT vulcanizates as compared to composites containing perlite and vermiculite. Halloysite nanotubes led to the best results, that, is, the lowest degree of chemical degradation of vulcanizates (approx. 38%), which is consistent with the literature. Other authors tentatively attributed this phenomenon to the infiltration of macromolecular polymeric chains through pores in nanotubes [83].

The surface of vulcanizates following exposure to n-heptane was examined using a stereoscopic microscope under 7× magnification (Opta-Tech series SK, Warsaw, Poland), with sample images presented in Figure 7.

### 2.9. Abrasion Resistance

The abrasion resistance of the fabricated vulcanizates was tested according to the procedures described in Section 3.10. In the case of this test, a deviation from the test in accordance with the standard EN 388:2016+A1:2018 [84] was used. Due to the high friction of the samples, the test was stopped after 500 rub cycles. The surface of the fabricated vulcanizates before tests and after 500 rub cycles was examined based on the acquired SEM images, with sample pictures presented in Figure 8.

The literature shows that the vulcanizate hardness increases with silicate concentration and with the diameter of filler particles due to higher cross-linking density resulting from polymer–filler interactions. Mostafa et al. observed greater abrasion-related mass loss for vulcanizates containing fillers (vs. unfilled ones) and for vulcanizates with higher filler concentrations [85].

## 3. Materials and Methods

### 3.1. Materials

In this work, butyl rubber, IIR (type: Butyl 206) with 2.3% mol. unsaturated bonds, a density of 0.91 g/mL, and Mooney viscosity of ML 1 + 8 (125 °C): 51 (delivered by ExxonMobil Chemicals&Specialties, Irving, TX, USA).

The curing system consisted of the following substances:-Sulfur (S) as a cross-linking agent, with a density of 1.8–2.36 g/mL (delivered by Chempur, Piekary Śląskie, Poland);-Zinc oxide (ZnO) as a cross-linking activator, with a density of 5.6 g/mL (delivered by Chempur, Piekary Śląskie, Poland);-Tetramethylthiuram disulfide (TMTD) as a cross-linking accelerator, with a density of 1.29 g/mL (delivered by Brenntag Polska Sp. z o. o., Kędzierzyn-Koźle, Polska);-Stearic acid (SA) as a cross-linking activator and dispersing agent, with a density of 0.94 g/mL (delivery by Chempur, Piekary Śląskie, Poland).

The following fillers were used:
-Silica (ARS), type: Arsil, with a density of 2.20 g/mL (delivered by Zakłady Chemiczne “Rudniki” S.A., Rudniki, Poland);-Vermiculite (VER), FlameHunter VE MIC, with a density of 0.85–1.00 g/mL, and an average particle size of 250–710 μm (>80%) (delivered by NYSA Chem^®^ Sp. z o. o., Wrocław, Poland);-Montmorillonite (MMT, NanoBent ZR), modified with a quaternary ammonium salt with two short- and two long-chain alkyl substituents, with an average particle size of 20–60 μm (56%), ≤20 μm (44%) and layer separation of 2.0–2.4 nm (delivered by ZGM “Zębiec”, Zębiec, Poland);-Perlite (PER), type: EP100F, with a density of 0.06–0.14 g/mL (delivered by Perlipol, Bełchatów, Poland);-Halloysite tubes (HNT), with a density of 2.53 g/mL (delivered by Sigma-Aldrich Chemie, Steinheim am Albuch, Germany).

### 3.2. Compounding and Vulcanization

The IIR composites were prepared using a two-roll mill (type: Laborwalzwerk, Krupp-Gruson, Magdeburg-Buckau, Germany) with a roll diameter of 200 mm and a roll length of 450 mm, at a roll temperature of 30–35 °C. The total time to create the composition was 5–8 min. First, the rubber was plasticized and the ingredients were incorporated in the following order: stearic acid, ZnO, filler, accelerator, and sulfur. The obtained rubber composites were stored separately in tightly closed foils at room temperature.

The produced mixes were vulcanized in hydraulic presses in appropriate metal molds. The vulcanization parameters were a temperature of 160 °C, a pressure of 150–180 bar, and a cure time of 30 min.

### 3.3. Characteristics of the Cross-Linking Process

The vulcanization process is characterized by determining the cure kinetics, and the equilibrium swelling. The cure kinetics of the IIR composites were determined using the Alpha Technologies (MDR 2000) oscillating disk rheometer (Alpha Technologies, Hudson, OH, USA) at 160 °C (ASTM D5289-17 standard [86]), which was employed to determine the following parameters: scorch time (t_02_); vulcanization time (t_90_); and minimal torque (T_min_); maximum torque increment (ΔT_max_), which is the difference between the torque after heating and minimal torque values. The degree of vulcanization (conversion) in the vulcametric studies was defined according to Formula (4) [87].
(4)$$\alpha =\frac{T_{t}-{\;T}_{0}}{T_{h}-{\;T}_{0}}$$
where T_t_ is the torque value at a given time during the vulcanization (in this case: T_15_), T_0_ is the torque value at time zero, T_h_ is the torque value at the end of vulcanization (in this case: T_30_).

The cure rate index (CRI) was designated according to Formula (5):(5)$$\mathrm{CRI}=\frac{100}{t_{90}-t_{02}}$$

Swelling behavior was assessed using toluene (according to ASTM D471 [88]). From each vulcanizate, four test pieces of 25–60 mg of different shapes were cut out, weighed using an electrical balance, and swollen in toluene until equilibrium was reached (for 72 h). After this time, the swollen samples were removed from toluene and washed with diethyl ether, and their weights were determined again. The samples were dried to a constant weight at a temperature of 50 °C and then reweighed.

Equilibrium volume swelling (Q_v_) was calculated using Formula (6):(6)$$Q_{v}={\;Q}_{w}\times \frac{d_{v}}{d_{s}}$$
where Q_w_ is the value of the equilibrium mass swelling (mg/mg), d_v_ is the vulcanizate density (g/mL), and d_s_ is the solvent density (g/mL).

Equilibrium weight swelling was calculated from Formula (7):(7)$$Q_{w}=\frac{m_{s}-m_{d}}{m_{d}^{*}}$$
where m_s_ is the swollen sample weight (mg), m_d_ is the dry sample weight (mg), and m_d_^*^ is the reduced sample weight (mg). The reduced sample weight was calculated from Formula (8):(8)$$m_{d}^{*}=m_{d}-m_{0}\cdot \frac{m_{m}}{m_{t}}$$
where m_0_ is the initial sample weight (mg), m_m_ is the mineral content in the blend (mg), and m_t_ is the total weight of the blend (mg).

Negative equilibrium weight swelling (*−*Q_w_), interpreted as the amount of leaching substances, was calculated from Formula (9):(9)$$-Q_{w}=\frac{m_{0}-m_{d}^{*}}{m_{0}}$$

The rubber volume fraction (V_R_) was calculated from Formula (10):(10)$$V_{R}=\frac{1}{1+Q_{v}}$$

The degree of cross-linking (α_c_) was determined using Formula (11):(11)$$\alpha_{c}=\frac{1}{Q_{v}}$$

### 3.4. Determination of Surface Morphology

The morphology of the vulcanizates was assessed using a scanning electron microscope (SEM). This was a Hitachi Tabletop Microscope TM-1000 (Tokyo, Japan) product. The preparation of the samples for measurement consisted of placing a double-sided self-adhesive foil onto a special table and gluing the testing sample to it. Then, a gold layer was applied to the prepared sample using the Cressington Sputter Coater 108 auto vacuum sputtering machine (Redding, CA, USA) at a pressure greater than 40 mbar for 60 s. The samples prepared in this way were placed into the scanning electron microscope chamber, and the measurement was performed.

### 3.5. Determination of Dynamic and Mechanical Properties

For the vulcanizates, the following properties were determined: strength properties, hysteresis losses and Mullins effect, tear resistance, hardness, elasticity, loss modules, and Payne effect.

Measurements of the tensile properties were carried out using a testing machine (Zwick1435/Roell GmbH & Co. KG, Ulm, Germany). The parameters determined from this test were stress at elongation of 100, 200, and 300% (S_e100_, S_e200_, S_e300_); tensile strength (TS_b_); and relative elongation at break (E_b_). Each property was determined for five samples. The test was conducted at a constant speed of 500 mm/min.

The hysteresis losses were determined using a testing machine (Zwick1435/Roell GmbH & Co. KG, Ulm, Germany). Each test was conducted for three samples, which were stretched five times to 200% elongation at a stretching speed of 500 mm/min, and the initial force was 0.1 N. The Mullins effect was determined according to Formula (12):(12)$$E_{M}=\frac{W_{1}-W_{5}}{W_{1}}\times 100\%$$
where W_1_ is the hysteresis loss at the first extension of the sample (N∙mm) and W_5_ is the hysteresis loss at the fifth extension of the sample (N∙mm).

The tear strength (T_s_) was tested in accordance with method A of the standard ISO 34-1:2022 [89] using a testing machine (Zwick1435/Roell GmbH & Co. KG, Ulm, Germany). Rectangular specimens with dimensions of 100 mm × 15 mm and a cut of 40 mm were used for the tests.

Hardness (HA) was tested on the Shore A scale using a Zwick/Roell hardness tester according to ISO-48-4:2018 [90]. Each test was performed ten times. The samples were in the shapes of cylinders, with diameters of 80 mm and heights of 6 mm.

The dynamic properties of the vulcanizates were determined by the minimum and maximum storage modulus (G′_min_, G′_max_), the maximum loss modulus (G′_max_), and the Payne effect (ΔG′, Formula (13)) at room temperature. The test was performed using the Ares G2 rotational rheometer (New Castle, UK) according to ISO 4664-1:2022 [91]. The tested samples, in the form of discs with dimensions of 25 mm × 2 mm, were placed between special measuring plates of the apparatus. The parameters that were used were as follows: a soak time of 10 s, an angular frequency of 10 rad/s, a logarithmic sweep with strain from 0.005 to 70% s, 20 points per decade, and an initial force of 5 N.
(13)$$\Delta G^{\prime }={G^{\prime }}_{\max}-{G^{\prime }}_{\min}$$
where G′_max_ is the maximum storage modulus (MPa) and G′_min_ is the minimum storage modulus (MPa).

### 3.6. Resistance to Thermo-Oxidative Aging

The thermal aging of the IIR vulcanizates was carried out in a forced circulating aging oven at 70 °C for 7 days. After conditioning at room temperature for 24 h, the changes of mechanical properties (stress at 100%, 200%, or 300% strain, tensile strength, elongation at break) were evaluated based on the aging factor (AF) according to Formula (14):(14)$$\mathrm{AF}=\frac{{\mathrm{TS}}_{b}^{\prime }\cdot E_{b}^{\prime }}{{\mathrm{TS}}_{b}\cdot E_{b}}$$
where ${\mathrm{TS}}_{b}^{\prime }$ is the tensile strength after thermo-oxidative aging (MPa), TS_b_ is the tensile strength before thermo-oxidative aging (MPa), $E_{b}^{\prime }$ is the elongation at break after thermo-oxidative aging (%), and E_b_ is the elongation at break before thermo-oxidative aging (%).

### 3.7. Determination of Hydrophobicity

The contact angle of the vulcanizates surface was determined using a goniometer from DataPhysics Instruments GmbH OCA 15EC (Filderstadt, Germany). The embedded drop method was used. At the beginning of the measurement, a drop of water with a volume of ~5 µL was placed on the surface of the vulcanizate using a Hamilton microsyringe. Then, using a special program, a photo of the drop was taken within 10 s so that the boundary between the surface and the drop was visible. The contact angle was measured by analysis in a computer program adapted for this study. A minimum of 5 drops were deposited into each sample and the average value of the contact angle was calculated.

### 3.8. Determination of Barrier Properties

The barrier properties were measured using a device that tests gas permeability using the manometric method. The measurement of barrier properties was based on a method that used pressure differences in measurement chambers on both sides of the tested sample. The apparatus consisted of a measuring cell in which the test sample was placed. The measuring cell was divided into two parts, i.e., an atmospheric pressure chamber and a low-pressure chamber. The test gas (air) was supplied to the chamber at atmospheric pressure. The low-pressure chamber contained a high-sensitivity sensor that measured pressure changes occurring in the measuring chamber as a result of gas permeating through the partition between both chambers containing the tested sample. A vacuum pump was connected to the low-pressure chamber, generating low pressure in the chamber (<10 Pa). From the results obtained, the gas transmission rate (GTR) was calculated according to Formula (15):(15)$$\mathrm{GTR}=\frac{V_{c}}{R\cdot T\cdot P_{u}\cdot A}\cdot \left(\mathrm{dp}/\mathrm{dt}\right)$$
where V_c_ is the volume of the low-pressure chamber (l), R is the gas constant (8.31 × 10^3^) [(l·Pa)/(K·mol)], T is the measurement temperature (K), P_u_ is the gas pressure in the high-pressure chamber (Pa), A is the area of gas permeation through the sample (m^2^), dp/dt is the pressure changes per unit of time (Pa/s).

The coefficient gas permeability (P) was determined according to Formula (16):(16)P=GTR·d
where d is the sample thickness (m).

### 3.9. Chemical Degradation

Chemical degradation studies for the obtained vulcanizates were carried out in accordance with the standard EN ISO 374-4:2019 [92]. Samples with a diameter of 20 mm were acclimatized at (23 ± 2 °C) for 24 h as per EN ISO 2231:1995 (PN-EN ISO 2231:1999) [93]. Then they were secured in glass vials containing 2 mL of heptane or methanol sealed with septa having a center hole 12 mm in diameter. The samples were placed under the septa. Vials prepared in this way were inverted to make sure that the test chemical was in direct contact with the vulcanizate surface for one hour.

Subsequently, the mechanical parameter of puncture force was determined in a comparative system, namely, for vulcanizates exposed and not exposed to the test chemicals (Figure 9); the latter served as reference values. Puncture force was measured at a probe advance rate of 100 mm/min (the initial distance between the puncture probe and the sample surface was 100 mm). The mean difference (*n* = 3) of the puncture force for samples exposed and not exposed to the test chemicals was expressed as a percentage.

### 3.10. Abrasion Resistance

Abrasion resistance was tested according to the standard PN-EN 388:2017-02 [94] with a modified maximum number of rubs. Samples with a diameter of 38 mm were acclimatized at a temperature of (23 ± 2) °C and a relative humidity of (50 ± 5)% for 24 h according to EN ISO 2231:1995 (PN-EN ISO 2231:1999). The mechanical parameter of resistance to cyclical abrasion was determined after 500 rubs executed at a force of (9 ± 0.2) kPa using a Martindale apparatus (James Heal, Sterling, VA, USA).

## 4. Conclusions

The effect of applied fillers on the IIR composite performance illustrates a radar chart (Figure 10).

Based on the research conducted, the following conclusions were drawn:The IIR composite filled with Arsil and vermiculite was characterized by the shortest cross-linking time (9 min) and the highest cure rate index (12.5 min^−1^).The highest degree of cross-linking (0.39) was achieved for IIR composites filled with Arsil or Arsil with perlite or halloysite tubes. The samples with the highest degree of cross-linking also obtained the highest mechanical strength.The tested vulcanizates did not show high tear resistance; however, the vulcanizates filled with Arsil and montmorillonite had the lowest tear resistance.The hardness of the tested compositions did not exceed 60 °ShA. The vulcanizates filled with Arsil and perlite were characterized by the highest hardness.Most of the tested compositions achieved an aging factor of approximately 0.6, which indicates good resistance to thermo-oxidative aging. The samples filled with Arsil and vermiculite or montmorillonite were characterized by the highest resistance to thermo-oxidative aging.Vulcanizates filled with Arsil achieved the lowest values of the Mullins effect, which may indicate the best degree of filler dispersion and the best vibration damping effect.The highest Payne effect was achieved for the IIR vulcanizate containing Arsil and perlite.The highest surface hydrophobicity (119°) was confirmed for the IIR vulcanizates with Arsil and montmorillonite.All tested samples showed high barrier properties because both the gas diffusion rate coefficient and the permeability coefficient reached low values.No changes were observed on the surface of the tested vulcanizates, such as shrinkage, chipping, peeling, or hardening, after contact with organic solvents of different polarity (methanol and n-heptane). Permanent deformation of the sample was visible after contact of all variants of vulcanizates with heptane.The sample retained chemical resistance in contact with methanol for 480 min. The presence of fillers did not reduce the protective parameter.In the case of contact of vulcanizates with n-heptane, the time taken for the liquid chemical substance to penetrate changed in the range from 210 to 380 min.Hour-long contact of a polar solvent (methanol) with each of the vulcanizates did not cause material degradation, while the presence of a non-polar solvent (heptane) worsened the mechanical parameters by up to 80%. However, the presence of fillers reduced the chemical degradation of vulcanizates (in the case of vulcanizates filled with Arsil and halloysite tubes by 40% compared to the vulcanizate without fillers).

## Figures and Tables

**Figure 1 molecules-29-01306-f001:**
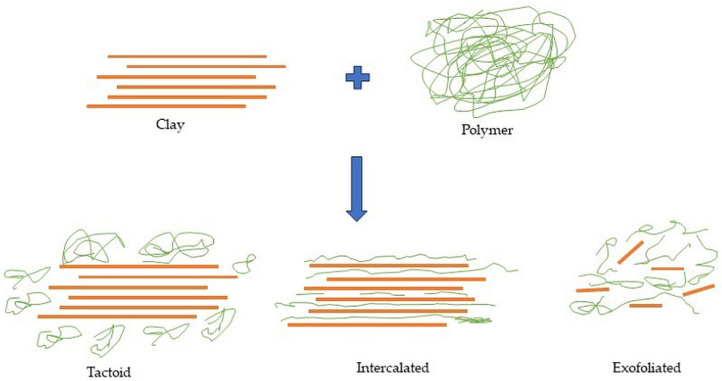
The possible morphologies of polymer/clay nanocomposites [38].

**Figure 2 molecules-29-01306-f002:**
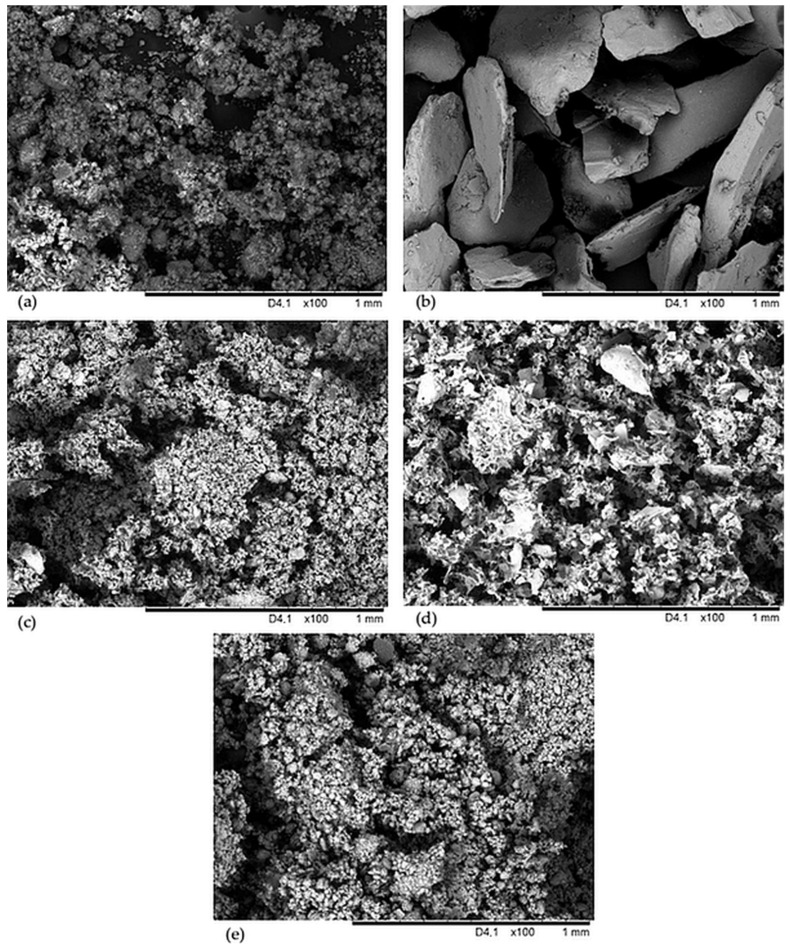
Scanning electron microscope (SEM) images of the fillers used in IIR composites: Arsil (**a**), vermiculite (**b**), montmorillonite (**c**), perlite (**d**), halloysite tubes (**e**).

**Figure 3 molecules-29-01306-f003:**
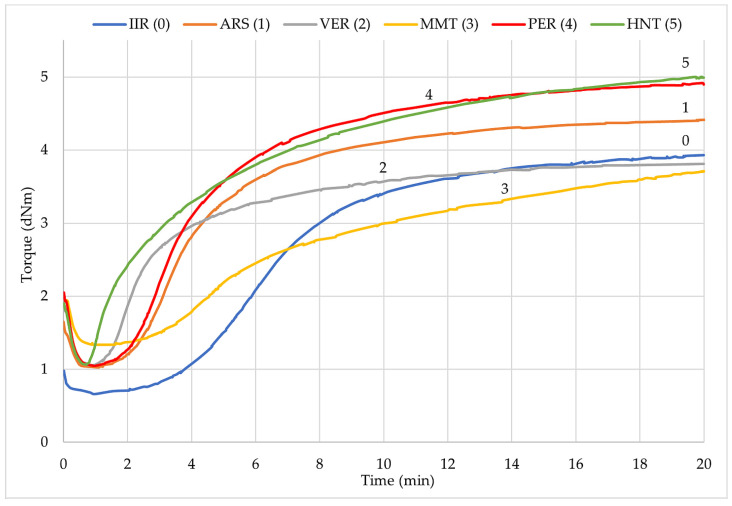
Vulcametric kinetics of IIR composites filled with phyllosilicates.

**Figure 4 molecules-29-01306-f004:**
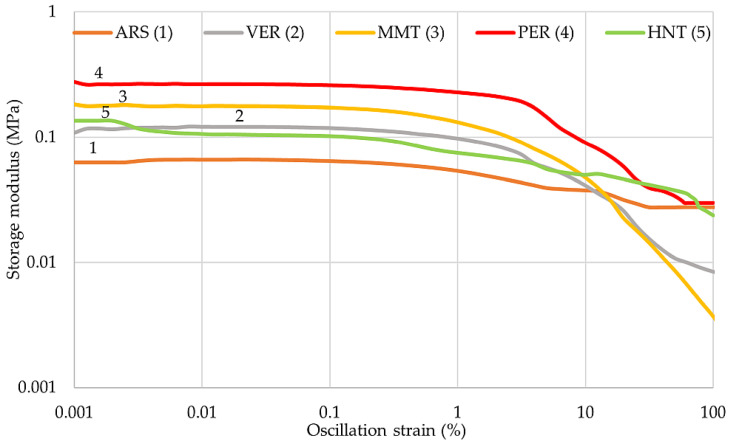
Relationship of storage modulus to oscillation strain of IIR composites filled with phyllosilicates.

**Figure 5 molecules-29-01306-f005:**
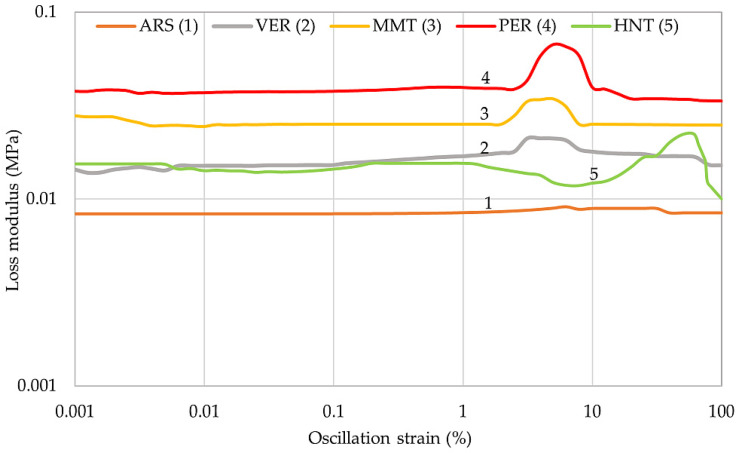
Relationship of loss modulus to oscillation strain of IIR composites filled with phyllosilicates.

**Figure 6 molecules-29-01306-f006:**
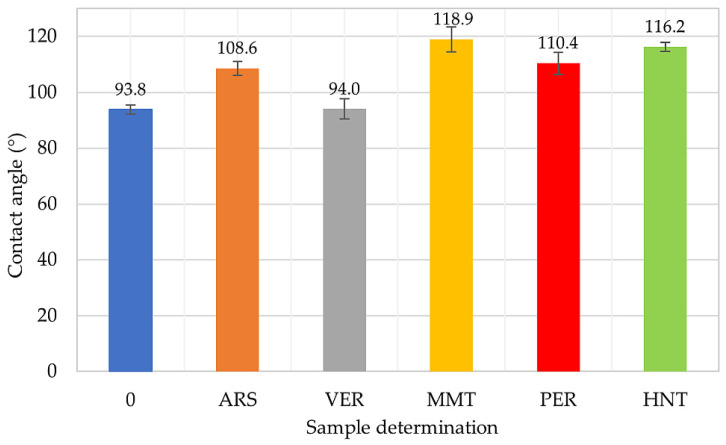
Contact angle values of IIR vulcanizates filled with phyllosilicates.

**Figure 7 molecules-29-01306-f007:**
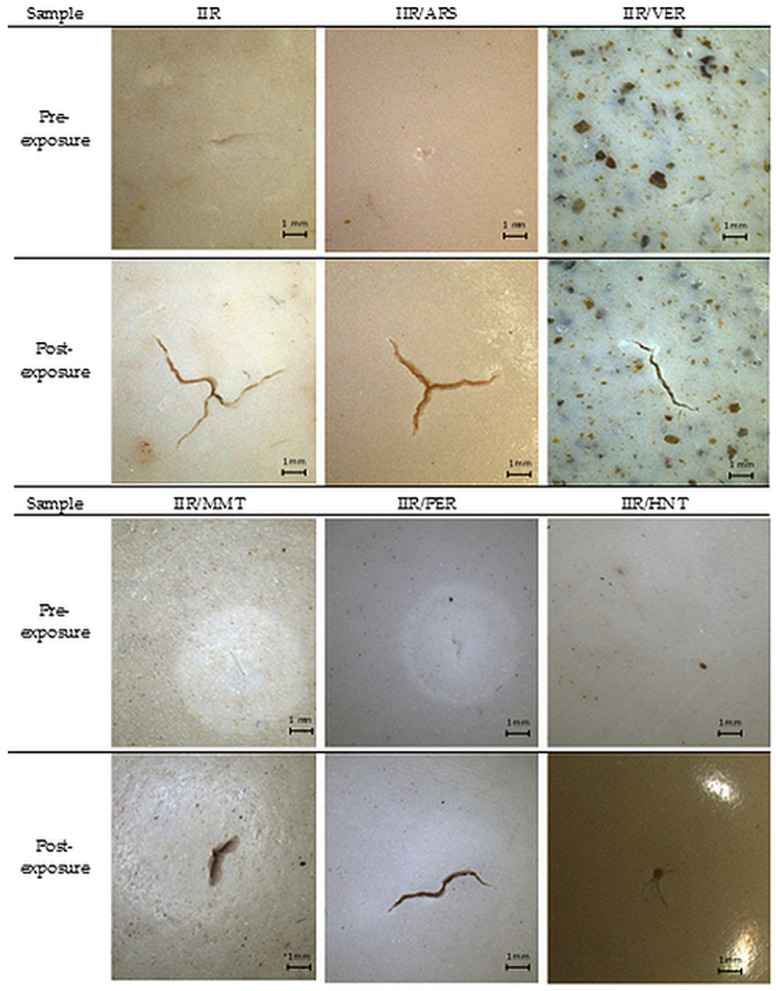
Stereoscopic microscope images of the surface of vulcanizates before and after n-heptane-induced chemical degradation.

**Figure 8 molecules-29-01306-f008:**
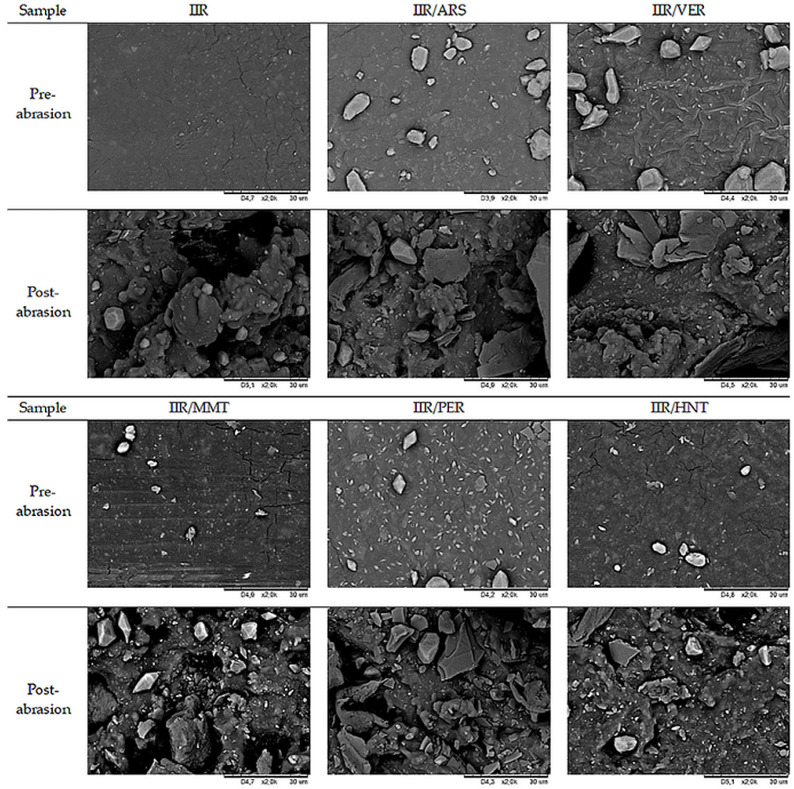
SEM images of the surface of fabricated vulcanizates before and after abrasion tests.

**Figure 9 molecules-29-01306-f009:**
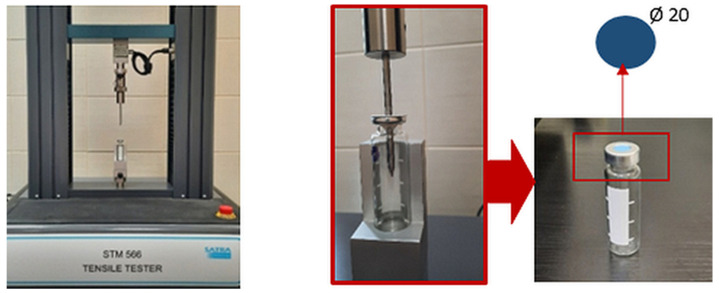
Measurement stand for determining chemical degradation according to EN ISO 374-4:2019 (Satra Technology, Kettering, United Kingdom).

**Figure 10 molecules-29-01306-f010:**
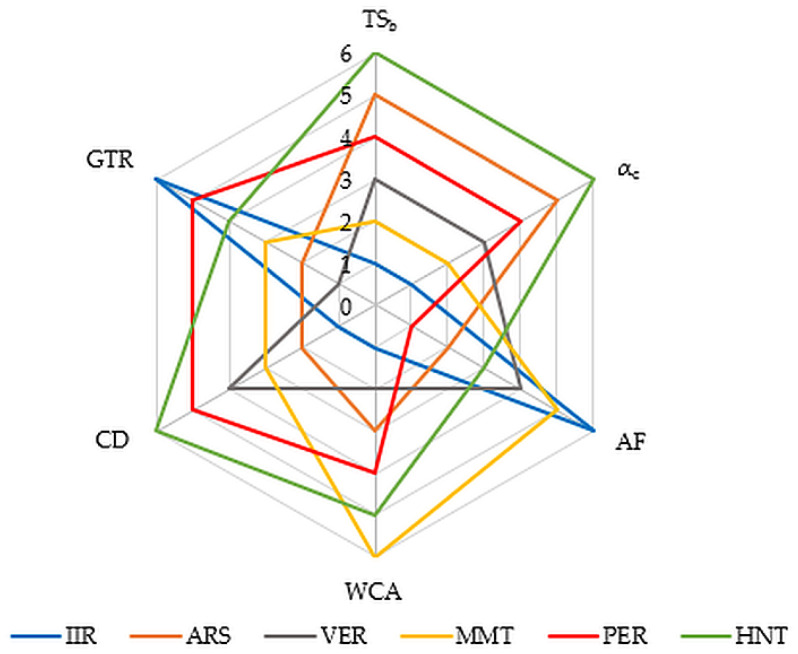
Comparison of properties of tested IIR vulcanizates filled with phyllosilicates (1—the weakest property; 6—the best property); TS_b_—tensile strength, α_c_—cross-linking degree, AF—aging factor, WCA—water contact angle, CD—chemical degradation, GTR—gas transmission rate.

**Table 1 molecules-29-01306-t001:** Compositions of the IIR mixes and vulcanizates.

Ingredient	Amount (phr)					
IIR	100	100	100	100	100	100
S	2	2	2	2	2	2
ZnO	5	5	5	5	5	5
TMTD	2	2	2	2	2	2
SA	2	2	2	2	2	2
ARS	-	10	10	10	10	10
VER	-	-	20	-	-	-
MMT	-	-	-	20	-	-
PER	-	-	-	-	20	-
HNT	-	-	-	-	-	20
Sample designation	0	ARS	VER	MMT	PER	HNT

IIR—butyl rubber, S—sulfur, ZnO—zinc oxide, TMTD—tetramethylthiuram disulfide, SA—stearic acid, ARS—silica, VER—vermiculite, MMT—montmorillonite, PER—perlite, HNT—halloysite tubes, phr—parts per hundred of rubber.

**Table 2 molecules-29-01306-t002:** Vulcametric parameters of IIR composites filled with silica and phyllosilicates.

Properties	Sample Designation					
	0	ARS	VER	MMT	PER	HNT
T_min_ (dNm)	0.66	1.03	1.06	1.34	1.05	1.07
ΔT_max_ (dNm)	3.27	3.44	2.75	2.37	3.87	3.93
α	0.95	0.96	0.97	0.84	0.96	0.94
t_02_ (min)	4.0	2.0	1.2	2.7	1.9	0.9
t_90_ (min)	12.0	10.4	9.2	16.0	10.3	12.3
CRI (1/min)	12.5	11.9	12.5	7.5	11.9	8.8
Q_v_ (mL/mL)	3.54 ± 0.06	2.58 ± 0.04	3.11 ± 0.06	3.79 ± 0.21	2.62 ± 0.05	2.58 ± 0.03
−Q_w_ (mg/mg)	0.27 ± 0.01	0.14 ± 0.01	0.27 ± 0.01	0.27 ± 0.01	0.26 ± 0.01	0.26 ± 0.01
V_R_	0.212 ± 0.002	0.280 ± 0.003	0.243 ± 0.003	0.209 ± 0.007	0.275 ± 0.003	0.279 ± 0.002
α_c_	0.28 ± 0.02	0.39 ± 0.03	0.32 ± 0.03	0.26 ± 0.02	0.38 ± 0.04	0.39 ± 0.03

T_min_—minimal rheometric torque; ΔT_max_—maximum rheometric torque increment; α—degree of conversion (vulcanization); t_02_—scorch time; t_90_—optimal vulcanization time; CRI—cure rate index; Q_v_—equilibrium volume swelling in toluene; −Q_w_—content of the eluted fraction in toluene; V_R_—volume fraction of rubber in swollen material in toluene; α_c_—degree of cross-linking.

**Table 3 molecules-29-01306-t003:** Strength properties of IIR vulcanizates filled with phyllosilicates before and after the thermo-oxidative aging process.

Properties	Sample Designation					
	0	ARS	VER	MMT	PER	HNT
S_e100_ (MPa)	0.56 ± 0.03	0.62 ± 0.03	0.48 ± 0.03	0.56 ± 0.01	0.66 ± 0.02	0.68 ± 0.01
S_e200_ (MPa)	0.83 ± 0.02	0.88 ± 0.05	0.68 ± 0.04	0.76 ± 0.02	0.90 ± 0.02	0.96 ± 0.02
S_e300_ (MPa)	1.14 ± 0.02	1.18 ± 0.08	0.88 ± 0.04	0.96 ± 0.03	1.18 ± 0.03	1.27 ± 0.03
TS_b_ (MPa)	2.5 ± 0.0	11.1 ± 1.1	7.9 ± 0.6	5.3 ± 0.4	11.0 ± 0.4	13.4 ± 1.0
E_b_ (%)	590 ± 57	1260 ± 56	1277 ± 19	1286 ± 1	1289 ± 2	1287 ± 2
S’_e100_ (MPa)	0.58 ± 0.08	0.62 ± 0.06	0.66 ± 0.02	0.67 ± 0.04	0.86 ± 0.04	0.82 ± 0.05
S’_e200_ (MPa)	0.87 ± 0.03	0.88 ± 0.06	0.90 ± 0.02	0.89 ± 0.05	1.12 ± 0.05	1.20 ± 0.09
S’_e300_ (MPa)	1.22 ± 0.09	1.16 ± 0.05	1.11 ± 0.02	1.12 ± 0.07	1.40 ± 0.08	1.56 ± 0.14
$TS_{b}^{\prime }$ (MPa)	2.3 ± 0.1	10.5 ± 1.4	8.7 ± 0.4	5.7 ± 0.4	6.6 ± 0.6	13.0 ± 0.7
$E_{b}^{\prime }$ (%)	558 ± 27	634 ± 34	732 ± 2	802 ± 24	673 ± 10	733 ± 9
AF (-)	0.85	0.48	0.63	0.67	0.31	0.55

S_e100_, S_e200_, and S_e300_—stress at elongation of 100%, 200%, and 300%, respectively; TS_b_—tensile strength; E_b_—elongation at break; S’_e100_, S’_e200_, and S’_e300_—stress at elongation of 100%, 200%, and 300%, respectively, after thermo-oxidative aging; ${\mathrm{TS}}_{b}^{\prime }$—tensile strength after thermo-oxidative aging; $E_{b}^{\prime }$—elongation at break after thermo-oxidative aging; AF—aging factor.

**Table 4 molecules-29-01306-t004:** Other mechanical properties of IIR vulcanizates filled with phyllosilicates.

Properties	Sample Designation					
	0	ARS	VER	MMT	PER	HNT
HA (°ShA)	42.8 ± 1.6	49.6 ± 2.4	50.8 ± 2.3	53.7 ± 1.5	57.6 ± 1.2	56.2 ± 2.4
T_s_ (N/mm)	2.29 ± 0.06	3.81 ± 0.53	4.45 ± 0.39	5.46 ± 0.25	3.62 ± 0.33	3.87 ± 0.32
ΔW_1_ (N·mm)	-	39.9	43.2	50.1	51.1	51.2
ΔW_5_ (N·mm)	-	27.4	25.1	24.1	26.4	29.9
E_M_ (%)	-	13.5	23.3	32.9	26.7	20.7

HA—hardness; T_s_—tear resistance; ΔW_1_, ΔW_5_—hysteresis losses during the first and fifth sample stretching cycles, E_M_—Mullins effect.

**Table 5 molecules-29-01306-t005:** Dynamic properties of IIR vulcanizates filled with phyllosilicates.

Properties	Sample Designation				
	ARS	VER	MMT	PER	HNT
G′_max_ (MPa)	0.066	0.122	0.183	0.278	0.172
G″_max_ (MPa)	0.012	0.022	0.033	0.067	0.022
ΔG′ (MPa)	0.040	0.116	0.181	0.256	0.144

G′_max_—maximum storage modulus; G″_max_—maximum loss modulus; ΔG′—Payne effect.

**Table 6 molecules-29-01306-t006:** Barrier properties of IIR vulcanizates filled with phyllosilicates.

Properties	Sample Designation					
	0	ARS	VER	MMT	PER	HNT
GTR ($\frac{\mathrm{mol}}{m^{2}\cdot s\cdot \mathrm{Pa}}$)	2.27 × 10^−9^	1.10 × 10^−8^	1.12 × 10^−8^	1.10 × 10^−8^	9.59 × 10^−9^	1.06 × 10^−8^
P ($\frac{\mathrm{mol}}{m\cdot \;s\cdot \mathrm{Pa}}$)	2.62 × 10^−12^	1.10 × 10^−8^	1.21× 10^−11^	1.54 × 10^−11^	1.09 × 10^−11^	1.09 × 10^−11^

GTR—gas transmission rate; P—coefficient gas permeability.

**Table 7 molecules-29-01306-t007:** Test results for permeation by liquid chemicals.

Properties	Test Substance	Sample Designation					
		0	ARS	VER	MMT	PER	HNT
Breakthrough time (min)	methanol	>480					
	n-heptane	249	280	210	380	289	295

**Table 8 molecules-29-01306-t008:** Test results for methanol-induced chemical degradation.

Properties	Test Substance	Sample Designation					
		0	ARS	VER	MMT	PER	HNT
Results for methanol-induced chemical degradation							
Mean puncture force (N)	Pre-exposure	51.51	58.65	32.80	27.92	37.68	41.68
	Post-exposure	50.96	48.84	35.23	28.06	39.00	41.39
Sample degradation (%)		1.07	16.74	−7.39	−0.49	−3.49	0.70
Results for n-heptane-induced chemical degradation							
Mean puncture force (N)	Pre-exposure	51.51	58.65	32.80	27.92	37.68	41.68
	Post-exposure	10.60	21.79	14.43	10.53	17.24	25.48
Sample degradation (%)		79.42	62.84	56.01	62.30	54.24	38.88

## Data Availability

Data are contained within the article.

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
