# Peer review of "The New Elastomeric Compounds Made of Butyl Rubber Filled with Phyllosilicates, Characterized by Increased Barrier Properties and Hydrophobicity and Reduced Chemical Degradation"

_molecules, 2024, doi:10.3390/molecules29061306_

Round 1
Reviewer 1 Report (Previous Reviewer 1)
Comments and Suggestions for Authors
No further comments.
Reviewer 2 Report (Previous Reviewer 2)
Comments and Suggestions for Authors
The authors have addressed all of my comments and I suggest publishing it in Molecules.
Reviewer 3 Report (Previous Reviewer 4)
Comments and Suggestions for Authors
This paper is well revised and can be accepted in the present form.
This manuscript is a resubmission of an earlier submission. The following is a list of the peer review reports and author responses from that submission.
Round 1
Reviewer 1 Report
Comments and Suggestions for Authors
This manuscript described the effect of different fillers on the properties of butyl rubber. However, the authors did not provide enough introduction background and justification for this specific research. In addition, information described in this manuscript is not enough to support the conclusions. Therefore, the publication of this manuscript in the journal is not recommended. Specifically, some recommendations are listed below to further improve the manuscript.
1. The introduction section needs more information to justify the study, especially the background of using different fillers in rubbers.
2. What are the major properties of the filled rubbers concerned in this study? These should be the research focus.
3. It seems that the sections 2 and 3 are misplaced.
4. The introduction to contact angle and hydrophobicity seems not necessary.
5. How to justify the particle size effect of various fillers?
Reviewer 2 Report
Comments and Suggestions for Authors
In this manuscript, the authors used silica and phyllosilicates to fill butyl rubber (IIR) and obtained a new elastomeric material with excellent properties. The effect of different fillers on the cross-linking properties, mechanical properties, dynamic properties, hydrophobicity, barrier properties, resistance to permeation, chemical degradation properties and abrasion resistance of butyl rubber was explored. The paper provides very interesting data but it still needs a considerable revision to be acceptable for the Molecules. The authors should also clarify/correct the points listed below. All points should be included in the manuscript.
1. P2 L53
Young’s angle needs to be explained.
2. P4 L142-145
Here you need to add some references to substantiate this statement.
3. P4 L146, P15 L460, P19 L642
These places have incorrect title serial numbers, but not just these, so please double check and verify.
4. P4 L159
HNT should be given its full name when it appears the first time in the manuscript.
5. P7 L248 P8 L256
The HNT serial number on the chart is incorrect and needs to be corrected.
6. P14 L446
Better to indicate the microscope scale on the drawing.
7. P19 L643
It is recommended that a radar chart be added to the conclusion section to illustrate the effect of different fillers on performance.
Reviewer 3 Report
Comments and Suggestions for Authors
Reviewer 4 Report
Comments and Suggestions for Authors
In this paper, the authors reported on novel composites made by combining IIR and phyllosilicates. While the work is interesting and the authors have invested significant effort, I observed that they did not analyze the reasons behind the composites exhibiting different enhanced properties. The manuscript presents results without providing explanations, which is insufficient for a research article. Additionally, the introduction part should address the recent research on IIR composites. Also, the representation of various figures should be enhanced. Therefore, I recommend that the authors resubmit the article after addressing the aforementioned points.